# EEG-GRAPH: A Factor-Graph-Based Model for Capturing Spatial, Temporal, and Observational Relationships in Electroencephalograms

Yogatheesan Varatharajah *      Min Jin Chong*      Krishnakant Saboo*      Brent Berry[†]

Benjamin Brinkmann[†]      Gregory Worrell[†]      Ravishankar Iyer*

## Abstract

This paper presents a probabilistic-graphical model that can be used to infer characteristics of instantaneous brain activity by jointly analyzing spatial and temporal dependencies observed in electroencephalograms (EEG). Specifically, we describe a factor-graph-based model with customized factor-functions defined based on domain knowledge, to infer pathologic brain activity with the goal of identifying seizure-generating brain regions in epilepsy patients. We utilize an inference technique based on the graph-cut algorithm to exactly solve graph inference in polynomial time. We validate the model by using clinically collected intracranial EEG data from 29 epilepsy patients to show that the model correctly identifies seizure-generating brain regions. Our results indicate that our model outperforms two conventional approaches used for seizure-onset localization (5–7% better AUC: **0.72**, 0.67, 0.65) and that the proposed inference technique provides 3–10% gain in AUC (**0.72**, 0.62, 0.69) compared to sampling-based alternatives.

## 1 Introduction

Studying the neurophysiological processes within the brain is an important step toward understanding the human brain. Techniques such as electroencephalography are exceptional tools for studying the neurophysiological processes, because of their high temporal and spatial resolution. An electroencephalogram (EEG) typically contains several types of rhythms and discrete neurophysiological events that describe instantaneous brain activity. On the other hand, the neural activity taking place in a brain region is very likely dependent on activities that took place in the same region at previous time instances. Furthermore, some EEG channels show inter-channel correlation due to their spatial arrangement [1]. Those three characteristics are related, respectively, to the *observational, temporal*, and *spatial* dependencies observed in time-series EEG signals.

The majority of the literature focuses on identifying and developing detectors for features relating to the different rhythms and discrete neurophysiological events in the EEG signal [2]. Some effort has been made to understand the inter-channel correlations [3] and temporal dependencies [4] observed in EEG. Despite these separate efforts, very little effort has been made to combine those dependencies into a single model. Since those dependencies possess complementary information, using only one of them generally results in poor understanding of the underlying neurophysiological phenomena. Hence, a unified framework that jointly captures all three dependencies in EEG, addresses an important research problem in electrophysiology. In this paper, we describe a graphical-model-based approach to capture all three dependencies, and we analyze its efficacy by applying it to a critical problem in clinical neurology.

[†]Department of Neurology, Mayo Clinic, Rochester, Minnesota 55904. Email: {Berry.Brent, Brinkmann.Benjamin, Worrell.Gregory}@mayo.edu

Graphical models in general are useful for representing dependencies between random variables. Factor graphs are a specific type of graphical models that have random variables and factor functions as the vertices in the graph [5]. A factor function is used to describe the relationship between two or more random variables in the graph. Factor graphs are particularly useful when custom definitions of the dependencies, such as in our case, need to be encoded in the graph. Hence, we have chosen to adopt a factor graph model to represent the three kinds of dependencies described previously. These dependencies are represented via three different factor functions, namely *observational*, *spatial*, and *temporal* factor functions. We assess the applicability of this model in localization of seizure onset zones (SOZ), which is a critical step in treating patients with epilepsy [6]. In particular, our model is utilized to isolate those neural events in EEG that are associated with the SOZ, and are eventually used to deduce the location of the SOZ. However, in a general setting, with appropriate definitions of factor functions, one can utilize our model to describe other neural events of interest (e.g., events related to behavioral states or memory processing). Major contributions of our work are the following.

1. A framework based on factor graphs that jointly represents instantaneous observation-based, temporal, and spatial dependencies in EEG. This is the first attempt to combine these three aspects into a single model in the context of EEG analysis.

2. A lightweight and exact graph inference technique based on customized definitions of factor functions. Exact graph inference is typically intractable in most graphical model representations because of exponentially growing state spaces.

3. A markedly improved technique for localizing SOZ based on the factor-graph-based model developed in this paper. Existing approaches utilize only the observations made in the EEG to determine the SOZ and do not utilize spatial and temporal dependencies.

Our study establishes the feasibility of the factor-graph-based model and demonstrates its application in SOZ localization on a real EEG dataset collected from epilepsy patients who underwent epilepsy surgery. Our results indicate that utilizing the spatial and temporal dependencies in addition to observations made in the EEG provides a 5–7% improvement in the AUC (**0.72**, 0.67, 0.65) and outperforms alternative approaches utilized for SOZ localization. Furthermore, our experiments demonstrate that the lightweight graph inference technique provides a considerable improvement (3–10%) in SOZ localization compared to sampling-based alternatives (AUC: **0.72**, 0.62, 0.69).

## 2 Related work

Identifying features (or biomarkers) that describe underlying neurophysiological phenomena has been a major focus of research in the EEG literature [2]. Spectral features [7], interictal spikes [8], high-frequency oscillations [2], and phase-amplitude coupling [4] are some of the widely used features. Although feature identification is an important step in any electrophysiologic study, features alone often cannot completely describe the underlying physiological phenomena. Researchers have also looked at spatial connectivity between EEG channels as means of describing neurophysiological activities [3]. In recent times, because of the availability of long-term EEG recordings, understanding of the temporal dependencies within various brain activities has also advanced significantly [4]. A recent attempt at combining spatial and temporal constraints has shown promise despite lacking comprehensive validation [9]. Regardless, a throughly validated and general model that captures all the factors, and is applicable to a variety of problems has not, to our knowledge, been proposed in the EEG literature. Since the three factors are complementary to each other, a model that jointly represents them addresses an important research gap in the field of electrophysiology.

Graphical models have been widely used in medical informatics [10], intrusion detection [11], social network modeling [12], and many other areas. Although factor graphs are applicable in all these settings, their applications in practice are still very much dependent on problem-specific custom definitions of factor functions. Nevertheless, with some level of customization, our work provides a general framework to describe the different dependencies observed in EEG signals. A similar framework for emotion prediction is described in Moodcast [12], for which the authors used a factor graph model to describe the influences of historical information, other users, and dynamic status to predict a user's emotions in a social network setting. Although our factor functions are derived in a similar fashion, we show that graph inference can be performed exactly using the proposed lightweight algorithm, and that it outperforms the sampling-based inference method utilized in Moodcast. Our

algorithm for inference was inspired by [13], in which the authors used an energy-minimization-based approach for performing exact graph inference in a Markov random field-based model.

## 3 Model description

Here we provide a mathematical description of the model and the inference procedure. In a nutshell, we are interested in inferring the presence of a neurophysiological phenomenon of interest by observing rhythms and discrete events (referred to as *observations*) present in the EEG, and by utilizing their spatial and temporal patterns as represented by a probabilistic graph. Since the generality of our model relies on the ability to customize the definitions of specific dependencies described by the model, we have adopted a factor-graph-based setting to represent our model.

**Definitions:** Suppose that EEG data of a subject are recorded through $M$ channels. Initially, the data is discretized by dividing the recording duration into $N$ epochs. We represent the interactions between the channels at an epoch $n$ as a dynamic graph $G_n = (V, E_n)$, where $V$ is the set of $|V| = M$ channels and $E_n \subset V \times V$ is the set of undirected links between channels. The state of a channel $k$ in the $n^{\text{th}}$ epoch is denoted by $Y_n(k)$, which might represent a phenomenon of interest. For example, in the case of SOZ localization, the state might be a binary value representing whether the $k^{\text{th}}$ channel in the $n^{\text{th}}$ epoch exhibits a *SOZ-likely* phenomenon. We also use $Y_n$ to denote the states of all the channels at epoch $n$, and use $\mathcal{Y}$ to denote the set of all possible values that $Y_n(k)$ can take. We refer to the EEG rhythm or discrete event present in the EEG as *observations* and use $X_n(k)$ to denote the *observation* present in the $n^{\text{th}}$ epoch of the $k^{\text{th}}$ channel. Depending on the number of rhythms and/or events, $X_n(k)$ could be a scalar or vector random variable. The *observations* made in all the channels at epoch $n$ are denoted by $X_n$.

**Inference:** Given a dynamic network $G_n$, and the *observations* $X_n$, our goal is to infer the states of the channels at epoch $n$, i.e., $Y_n$. In our approach, we derive the inference model using a factor graph with factor functions defined as shown in Table 1. The factor functions are defined using exponential relationships so that they attain their maximum values when the exponents are zero, and exponentially decay otherwise. All factor functions range in $[0, 1]$.

Table 1: Factor functions used in our EEG model and their descriptions, definitions, and notations.

| Function | Description | Defnition | Notations |
|---|---|---|---|
| Observational: $f(Y_n(k), \phi(X_n(k)))$ | Measures the direct contribution of the observations made in a channel to the phenomenon of interest. | $e^{-(Y_n(k) - \phi(X_n(k)))^2}$ | $\phi : X \to \mathcal{Y}$ is a mapping from the observations to the phenomenon of interest. In general, it is not an accurate map, because it is based on observations alone. |
| Spatial: $g(Y_n(k), Y_n(l))$ | Measures the correlation between the states of two channels at the same epoch. | $e^{-\frac{1}{d_{kl}^2}(Y_n(k) - Y_n(l))^2}$ | $d_{kl}$ denotes the physical distance between electrodes (or channels) $k$ and $l$. |
| Temporal: $h(Y_n(k), \Omega_{n-1}(k))$ | Measures the correlation between a channel's current state and its previous states. | $e^{-\left(Y_n(k) - \Omega_{n-1}(k)\right)^2}$ | $\Omega_{n-1}(k)$ is a function of all previous states of channel $k$. E.g., $\Omega_{n-1}(k) = \frac{\sum_{i=1}^{n-1} Y_i(k)}{n-1}$ |

With these definitions, the state of a channel is spatially related to the states of every other channel, temporally related to a function of all its previous states, and, at the same time, explained by the current observation of the channel. These dependencies and the factor functions that represent them are illustrated in Fig. 1a and 1b respectively. (Note that Fig. 1b illustrates only the factor functions related to Channel 1 and that similar factor functions exist for other channels although they are not shown in the figure.) Provided with that information, for a particular state vector $Y$, we can write $P(Y|G_n)$ as in Eq. 1, where $Z$ is a normalizing factor. In general, it is infeasible to find the normalizing constant $Z$, because it would require exploration of the space $|\mathcal{Y}|^M$.

$$P(Y|G_n) = \frac{1}{Z} \prod_{k=1}^{M} \left[ \prod_{i \neq k} g\left(Y(k), Y(i)\right) \times f\left(Y(k), \phi(X_n(k))\right) \times h\left(Y(k), \Omega_{n-1}(k)\right) \right] \quad (1)$$

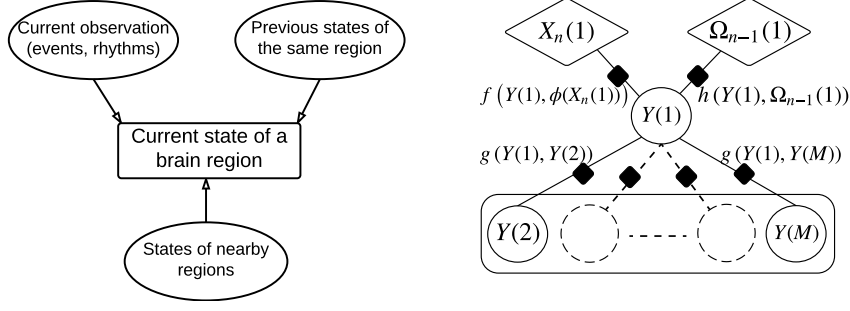

(a) Factors that explain the state of a brain region.   (b) Dependencies as factor functions.

Figure 1: The dependencies observed in brain activity and a representative factor graph model.

Therefore, we define the following predictive function (Eq. 2) for inferring $Y_n$ with the highest likelihood per Eq. 1.

$$Y_n = \underset{Y \in \mathcal{Y}^M}{\arg\max} \prod_{k=1}^{M} \left[ \prod_{i \neq k} g\left(Y(k), Y(i)\right) \times f\left(Y(k), \phi(X_n(k))\right) \times h\left(Y(k), \Omega_{n-1}(k)\right) \right] \quad (2)$$

Still, finding a $Y$ that maximizes this objective function involves a discrete optimization over the space $|\mathcal{Y}|^M$. A brute-force approach to finding an exact solution is infeasible when $M$ is large. Several methods, such as junction trees [14], belief propagation [15], and sampling-based methods such as Markov Chain Monte Carlo (MCMC) [16, 17], have been proposed to find approximate solutions. However, we show that this can be calculated exactly when the aforementioned definitions of the factor functions are utilized. We can rewrite Eq. 2 using the definitions in Table 1 as follows.

$$Y_n = \underset{Y \in \mathcal{Y}^M}{\arg\max} \prod_{k=1}^{M} \left[ \prod_{l \neq k} e^{-\frac{1}{d_{kl}^2}(Y(k)-Y(l))^2} \times e^{-(Y(k)-\phi(X_n(k)))^2} \times e^{-(Y(k)-\Omega_{n-1}(k))^2} \right] \quad (3)$$

Now, representing the product terms as summations inside the exponent and using the facts that the exponential function is monotonically increasing and that maximizing a function is equivalent to minimizing the negative of that function, we can rewrite Eq. 3 as:

$$Y_n = \underset{Y \in \mathcal{Y}^M}{\arg\min} \sum_{k=1}^{M} \left[ \sum_{l \neq k} \frac{1}{d_{kl}^2}(Y(k)-Y(l))^2 + (Y(k)-\phi(X_n(k)))^2 + (Y(k)-\Omega_{n-1}(k))^2 \right] \quad (4)$$

Although the individual components in this objective function are solvable optimization problems, the combination of them makes it difficult to solve. However, the objective function resembles that of a standard graph energy minimization problem and hence can be solved using graph-cut algorithms [18]. In this paper, we describe a solution for minimizing this objective function when $|\mathcal{Y}| = 2$, i.e., the brain states are binary. Although that is a limitation, the majority of the brain state classification problems can be reduced to binary state cases when the time window of classification is appropriately chosen. Regardless, potential solutions for $|\mathcal{Y}| > 2$ are discussed in Section 6.

**Graph inference using min-cut for the binary state case:** We constructed the graph shown in Fig. 2a with two special nodes in addition to the EEG channels as vertices. The additional nodes function as *source* (marked by 1) and *sink* (marked by 0) nodes in the conventional min-cut/max-flow problem. Weights in this graph are assigned as follows:

- Every channel is connected with every other channel, and the link between channels $k$ and $l$ is assigned a weight of $\frac{1}{d_{kl}^2}(Y(k) - Y(l))^2$ based on the distance between them.
- Every channel is connected with the source node, and the link between channel $k$ and the source is assigned a weight of $(1 - \Omega_{n-1}(k))^2 + (1 - \phi(X_n(k)))^2$.
- Every channel is also connected with the sink node, and the link between channel $k$ and the sink is assigned a weight of $\Omega_{n-1}^2(k) + (\phi(X_n(k)))^2$.

**Proposition 1.** An optimal min-cut partitioning of the graph shown in Fig. 2a minimizes the objective function given in Eq. 4.

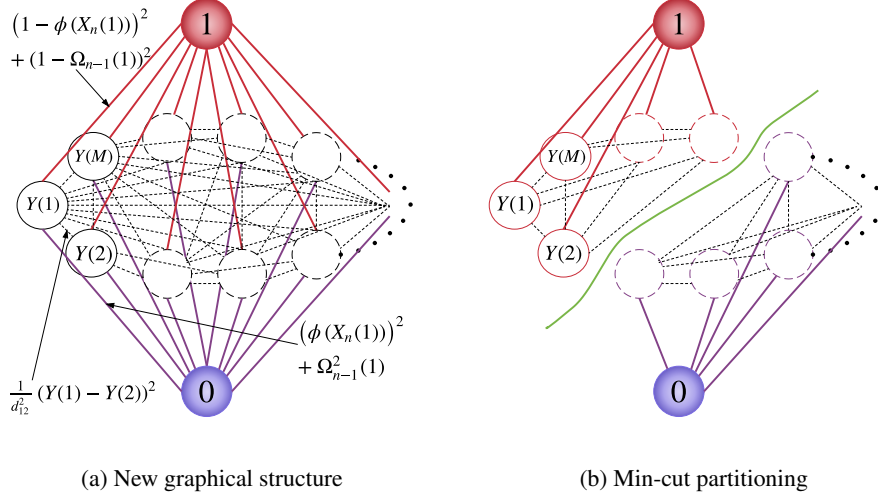

(a) New graphical structure  (b) Min-cut partitioning

Figure 2: Graph inference using the min-cut algorithm.

**Proof:** Suppose that we perform an arbitrary cut on the graph shown in Fig. 2a, resulting in two sets of vertices $\mathcal{S}$ and $\mathcal{T}$. The energy of the graph after the cut is performed is:

$$\mathcal{E}_{cut} = \sum_{k=1}^{M} \left[ (Y(k) - \Omega_{n-1}(k))^2 + (Y(k) - \phi(X_n(k)))^2 \right] + \sum_{k \in \mathcal{T}} \sum_{l \in \mathcal{S}} \left[ \frac{1}{d_{kl}^2} (Y(k) - Y(l))^2 \right]$$

It can be seen that, for the same partition of vertices, the objective function given in Eq. 4 attains the same quantity as $\mathcal{E}_{cut}$. Therefore, since the optimal min-cut partition minimizes the energy $\mathcal{E}_{cut}$, it minimizes the objective function given in Eq. 4.

Now suppose that we are given two sets of nodes $\{\mathcal{S}^*, \mathcal{T}^*\}$ as the optimal partitioning of the graph. Without loss of generality, let us assume that $\mathcal{S}^*$ contains the *source* and $\mathcal{T}^*$ contains the *sink*. Then, the other vertices in $\mathcal{S}^*$ and $\mathcal{T}^*$, are assigned 1 and 0 as their respective states to obtain the optimal $Y$ that minimizes the objective function given in Eq. 4.

## 4 Application of the model in seizure onset localization

**Background:** Epilepsy is a neurological disorder characterized by spontaneously occurring seizures. It affects roughly 1% of the world's population, and many do not respond to drug treatment [19]. Epilepsy surgery, which involves resection of a portion of the patient's brain, can reduce and often eliminate seizures [20]. The success of resective surgery depends on accurate localization of the seizure-onset zone [21]. The conventional practice is to identify the EEG channels that show the earliest seizure discharge via visual inspection of the EEG recorded during seizures, and to remove some tissue around these channels during the resective surgery. This method, despite being the current clinical standard, is very costly, time-consuming, and burdensome to the patients, as it requires a lengthy ICU stay so that an adequate number of seizures can be captured. One approach, which has recently become a widely researched topic, utilizes between-seizure (interictal) intracranial EEG (iEEG) recording to localize the seizure onset zones [22, 6]. This type of localization is preferable to the conventional method, as it does not require a lengthy ICU stay.

**Interictal SOZ identification methodology:** Like that of the conventional approach, the goal here is to identify a few channels that are likely to be in the SOZ. Channels situated directly on or close to a SOZ exhibit different forms of transient electrophysiologic events (or abnormal events) between seizures [23]. The frequency of such abnormal neural events plays a major role in determining the SOZ. However, capturing these abnormal neural events that occur in distinct locations of the brain alone is often not sufficient to establish an area in the brain as the SOZ. The reason is that insignificant artifacts present in the EEG may show characteristics of those abnormal events that are associated with SOZ (referred to as SOZ-likely events). In order to set apart the SOZ-likely events, their spatial and temporal patterns could be utilized. It is known that SOZ-likely events occur in a repetitive and spatially correlated fashion (i.e., neighboring channels exhibit such events at the same time) [6]. Hence, the factor-graph-based model described in Section 3 can be applied to capture and utilize the spatial and temporal correlations in isolating the SOZ-likely events.

**Identifying abnormal neural events:** Spectral characteristics of iEEG measured in the form of power-in-bands (PIB) features have been widely utilized to identify abnormal neural events [24, 6, 7]. In this paper, PIB features are extracted as spectral power in the frequency bands *Delta* (0–3 Hz), *Low-Theta* (3–6 Hz), *High-Theta* (6–9 Hz), *Alpha* (9–14 Hz), *Beta* (14–25 Hz), *Low-Gamma* (30–55 Hz), *High-Gamma* (65–115 Hz), and *Ripple* (125–150 Hz) and utilized to make observations from channels. As described in Section 3, a $\phi$ function is used to relate the observations to abnormal events. In Section 6, we evaluate different techniques for obtaining a mapping from extracted PIB features to the presence of an abnormal neural event. However, a mapping obtained using observations alone is not sufficient to deduce SOZ because in addition to SOZ-likely events, signal artifacts will also be captured by this mapping. This phenomenon is illustrated in Fig.3, in which PIB features show similar characteristics for the events related to both SOZ and non-SOZ. Therefore, we utilize the factor graph model presented in this paper to further filter the detected abnormal events based on their spatial and temporal patterns and isolate the SOZ-likely events.

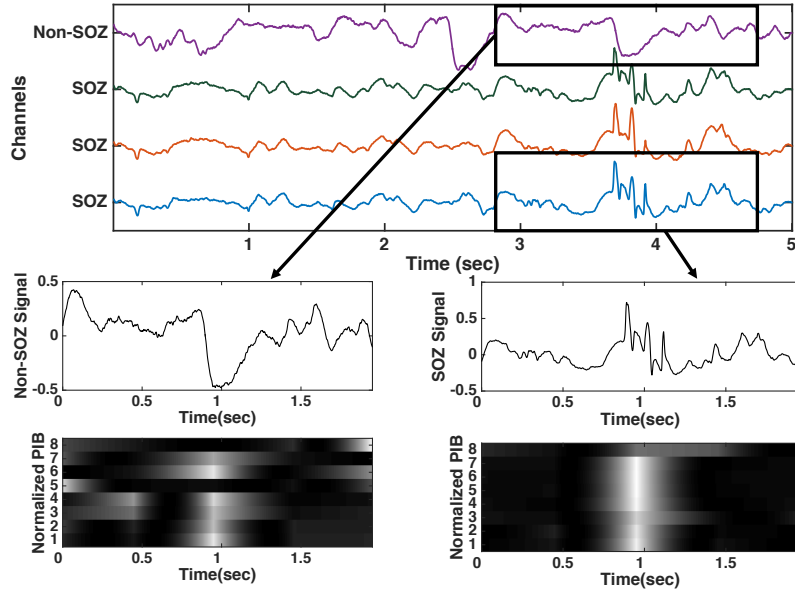

Figure 3: EEG events related to both SOZ and non-SOZ are captured by PIB features because they possess similar spectral characteristics.

**Spatial and temporal dependencies in SOZ localization:** Although artifacts show spectral characteristics similar to those of SOZ-likely events, unlike the latter, the former do not occur in a spatially correlated manner. This spatial correlation is measured with respect to the physical distances between the electrodes placed in the brain. Therefore, the same definition of the spatial factor function described in Section 3 is applicable. If a channel's observation is classified as an abnormal neural event and the spatial factor function attains a large value with an adjacent channel, it would mean that both channels likely show similar patterns of abnormalities which therefore must be SOZ-likely events. In addition, the SOZ-likely events show a repetitive pattern, which artifacts usually do not. In Section 3, we described the temporal correlation as a function of all previous states. As such, the temporal correlation here is established with the intuition that a channel that previously exhibited a large number of SOZ-likely events is likely to exhibit more because of the repetitive pattern. Hence, temporal correlation is measured as the correlation between the state of a channel and the observed frequency of SOZ-likely events in that channel until the previous epoch, i.e., $\Omega_{n-1}(k) = \frac{\sum_{i=1}^{n-1} Y_i(k)}{n-1}$. Therefore, when $\Omega_{n-1}(k)$ is close to 1 and the observation made from channel $k$ is classified as an abnormal neural event, the event is more likely to be a SOZ-likely event than an artifact.

## 5  Experiments

**Data:** The data used in this work are from a study approved by the Mayo Clinic Institutional Review Board. The dataset consists of iEEG recordings collected from 29 epilepsy patients. The iEEG sensors were surgically implanted in potentially epileptogenic regions in the brain. Patients were

implanted with different numbers of sensors, and they all had different SOZs. Ground truth (the true SOZ channels) was established from clinical reports and verified independently through visual inspection of the seizure iEEGs. During data collection, basic preprocessing was performed to remove line-noise and other forms of signal contamination from the data.

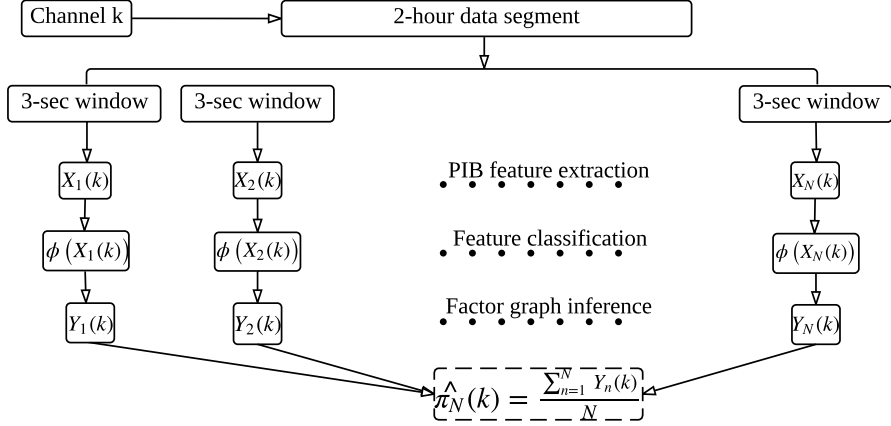

Figure 4: A flow diagram illustrating the SOZ determination process.

**Analytic scheme:** Two-hour between-seizure segments were chosen for each patient to represent a monitoring duration that could be achieved during surgery. The two-hour iEEG recordings were divided into non-overlapping three-second epochs. This epoch length was chosen because it would likely accommodate at least one abnormal neural event that could be associated with the SOZ [6]. Spectral domain features (PIB) were extracted in the 3-second epochs to capture abnormal neural events [6]. Based on the features extracted in a 3-second recording of a channel, a binary value $\phi(X_n(k)) \in \{0, 1\}$ was assigned to that channel, indicating whether or not an abnormal event was present. Section 6 provides a comparison of supervised and unsupervised techniques used to create this mapping. In the case of supervised techniques, a classification model was trained using the PIB features extracted from an existing corpus of manually annotated abnormal neural events. In the case of unsupervised techniques, channels were clustered into two groups based on the PIB features extracted during an epoch, and the cluster with the larger cluster center (measured as the Euclidean distance from the origin) was labeled as the abnormal cluster. Consequently, the respective epochs of those channels in the abnormal cluster were classified as abnormal neural events. The factor graph model was then used to filter the SOZ-likely events out of all the detected abnormal neural events. A factor graph is generated using the *observational, spatial*, and *temporal* factor functions described above specifically for this application. The best combination of states that minimizes the objective function given in Eq. 4, $Y_n$, is found by using the min-cut algorithm. In our approach, we used the *Boykov-Kolmogorov* algorithm [25] to obtain the optimal partition of the graph. The states $Y_n$ here are binary values and represent the presence or absence of SOZ-likely events in the channels. This process is repeated for all the 3-second epochs and the SOZ is deduced at the end using a maximum likelihood (ML) approach (described in the following). This whole process is illustrated in Fig. 4.

**Maximum likelihood SOZ deduction:** We model the occurrences of SOZ-likely events in channel $k$ as independent Bernoulli random variables with probability $\pi(k)$. Here, $\pi(k)$ denotes the true bias of the channel's being in SOZ. We estimate $\pi(k)$ using a maximum likelihood (ML) approach and use $\hat{\pi}(k)$ to denote the estimate. Each $Y_n(k)$ that results from the factor graph inference is treated as an outcome of a Bernoulli trial and the log-likelihood function after $N$ such trials is defined as:

$$\log(L(\pi(k))) = \log\left[\prod_{n=1}^{N} \pi(k)^{Y_n(k)} (1 - \pi(k))^{1 - Y_n(k)}\right] \tag{5}$$

An estimate for $\pi(k)$ that maximizes the above likelihood function (known as MLE, i.e., maximum likelihood estimate) after $N$ epochs is derived as $\hat{\pi}(k) = \frac{\sum_{n=1}^{N} Y_n(k)}{N}$.

**Evaluation:** The ML approach generates a likelihood probability for each channel $k$ for being in the SOZ. We compared these probabilities against the ground truth (binary values with 1 meaning

that the channel is in the SOZ and 0 otherwise) to generate the area under the ROC curve (AUC), sensitivity, specificity, precision, recall, and F1-score metrics. First, we evaluated a number of techniques for generating a mapping from the extracted PIB features to the presence of abnormal events. We evaluated three unsupervised approaches, namely k-means, spectral, and hierarchical clustering methods and two supervised approaches, namely support vector machine (SVM) and generalized linear model (GLM), for this task. Second, we evaluated the benefits of utilizing the min-cut algorithm for inferring instantaneous states. Here we compared our results using the min-cut algorithm against those of two sampling-based techniques [12]: MCMC with random sampling, and MCMC with sampling per prior distribution. Belief-propagation-based methods are not suitable here because our factor graph contains cycles [26]. Third, we compared our results against two recent solutions for interictal SOZ localization, including a summation approach [6] and a clustering approach [22]. In the summation approach, summation of the features of a channel normalized by the maximum feature summation was used as the likelihood of that channel's being in the SOZ. In the clustering approach, the features of all the channels during the whole 2-hour period were clustered into two classes by a k-means algorithm, and the cluster with the larger cluster mean was chosen as the abnormal cluster. For each channel, the fraction of all its features that were in the abnormal cluster was used as the likelihood of that channel being in the SOZ. Both of these approaches utilize only the observations and lack the additional information of the spatial and temporal correlations.

## 6 Results & discussion

Table 2 lists the results obtained for the experiments explained in Section 5, performed using a dataset containing non-seizure (interictal) iEEG data from 29 epilepsy patients. First, a comparison of supervised and unsupervised techniques for the mapping from PIB features to the presence of abnormal events was performed. The results indicate that using a k-means clustering approach for mapping PIB features to abnormal events is better than any other supervised or unsupervised approach, while other approaches also prove useful. Second, a comparison between sampling-based methods and the min-cut approach was performed for the task of graph inference. Our results indicate that utilizing the min-cut approach to infer instantaneous states is considerably better than a random-sampling-based MCMC approach (with a 10% higher AUC and 14% higher F1-score) and marginally better than an MCMC approach with sampling per a prior distribution (with a 3% higher AUC and a similar F1-score), when used with k-means algorithm for abnormal event classification. However, unlike this approach, our method does not require a prior distribution to sample from. Third, we show that our factor-graph-based model for interictal SOZ localization performs significantly better than either of the traditional approaches (with 5% and 7% higher AUCs) when used with k-means algorithm for abnormal event classification and min-cut algorithm for graph inference.

Table 2: Goodness-of-fit metrics obtained for unsupervised and supervised methods for PIB-to-abnormal-event mapping ($\phi$); sampling-based approaches for instantaneous state estimation; and conventional approaches utilized for interictal SOZ localization. ("FG/kmeans/min-cut" means that we utilized a factor-graph-based method, with a k-means clustering algorithm for mapping PIB featuers to abnormal neural events and the min-cut algorithm for performing graph inference.)

| Method | AUC | Sensitivity | Specificity | Precision | Recall | F1-score |
|---|---|---|---|---|---|---|
| **Evaluation: techniques for PIB to abnormal event mapping ($\phi$)** | | | | | | |
| FG/**kmeans**/min-cut | **0.72±0.03** | **0.74±0.03** | 0.61±0.02 | 0.39±0.05 | **0.74±0.03** | **0.46±0.04** |
| FG/**spectral**/min-cut | 0.68±0.03 | 0.60±0.07 | 0.48±0.05 | 0.31±0.05 | 0.60±0.07 | 0.36±0.05 |
| FG/**hierarch**/min-cut | 0.69±0.03 | 0.52±0.06 | 0.51±0.05 | 0.29±0.05 | 0.52±0.06 | 0.34±0.05 |
| FG/**svm**/min-cut | 0.71±0.03 | 0.68±0.06 | 0.54±0.05 | 0.36±0.05 | 0.68±0.06 | 0.43±0.05 |
| FG/**glm**/min-cut | 0.69±0.03 | 0.62±0.07 | 0.47±0.05 | 0.31±0.05 | 0.62±0.08 | 0.37±0.05 |
| **Evaluation: sampling vs. min-cut** | | | | | | |
| FG/kmeans/**Random** | 0.62±0.03 | 0.51±0.08 | 0.40±0.07 | 0.35±0.06 | 0.51±0.08 | 0.32±0.05 |
| FG/kmeans/**Prior** | 0.69±0.03 | 0.65±0.04 | 0.66±0.04 | 0.40±0.04 | 0.65±0.04 | **0.46±0.04** |
| **Evaluation: comparison against conventional approaches** | | | | | | |
| **Summation** | 0.67±0.04 | 0.59±0.05 | 0.67±0.03 | 0.38±0.05 | 0.59±0.05 | 0.43±0.05 |
| **Clustering** | 0.65±0.04 | 0.49±0.06 | **0.72±0.04** | **0.42±0.06** | 0.49±0.06 | 0.44±0.05 |

**Significance:** Overall, the factor-graph-based model with k-means clustering for abnormal event classification and the min-cut algorithm for instantaneous state inference outperforms all other methods for the application of interictal SOZ localization. Utilization of spatial and temporal factor functions improves the localization AUC by 5–7%, relative to pure observation-based approaches (summation and clustering). On the other hand, the runtime complexity of instantaneous state inference is greatly reduced by the min-cut approach. The complexity of a brute-force approach grows exponentially with the number of nodes in the graph, while the min-cut approach has a reasonable runtime complexity of $\mathcal{O}(|V||E|^2)$, where $|V|$ is the number of nodes and $|E|$ is the number of edges in the graph. Although sampling-based methods are able to provide approximate solutions with moderate complexity, the min-cut method provided superior performance in our experiments.

**Future work:** Significant domain knowledge is required to come up with manual definitions of graphical models, and in many situations, almost no domain knowledge is available. Hence, the manually defined factor-graphical model and associated factor functions are a potential limitation of our work, as a framework that automatically learns the graphical representation might result in a more generalizable model. Dynamic Bayesian networks [27] may provide a platform that can be used to learn dependencies from the data while allowing the types of dependencies we described. Another potential limitation of our work is the binary-brain-state assumption made while solving the graph energy minimization task. We surmise that extensions of the min-cut algorithm such as the one proposed in [28] are applicable for non-binary cases. In addition, we also believe that optimal weighting of the different factor functions could further improve localization accuracy and provide insights on the contributions of spatial, temporal, and observational relationships to a specific application that involves EEG signal analysis. We plan to investigate those in our future work.

## 7 Conclusion

We described a factor-graph-based model to encode observational, temporal, and spatial dependencies observed in EEG-based brain activity analysis. This model utilizes manually defined factor functions to represent the dependencies, which allowed us to derive a lightweight graph inference technique. This is a significant advancement in the field of electrophysiology because a general and comprehensively validated model that encodes different forms of dependencies in EEG does not exist at present. We validated our model for the application of interictal seizure onset zone (SOZ) and demonstrated the feasibility in a clinical setting. Our results indicate that our approach outperforms two widely used conventional approaches for the application of SOZ localization. In addition, the factor functions and the technology for exactly inferring the states described in this paper can be extended to other applications of factor graphs in fields such as medical diagnoses, social network analysis, and preemptive attack detection. Therefore, we assert that further investigation is necessary to understand the different usecases of this model.

**Acknowledgements:** This work was partly supported by National Science Foundation grants CNS-1337732 and CNS-1624790, National Institute of Health grants NINDS-U01-NS073557, NINDS-R01-NS92882, NHLBI-HL105355, and NINDS-UH2-NS095495-01, Mayo Clinic and Illinois Alliance Fellowships for Technology-based Healthcare Research and an IBM faculty award. We thank Subho Banerjee, Phuong Cao, Jenny Applequist, and the reviewers for their valuable feedback.

## Footnotes

*Department of Electrical and Computer Engineering, University of Illinois at Urbana-Champaign, Urbana, Illinois 61801. Email: {varatha2, mchong6, ksaboo2, rkiyer}@illinois.edu

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
