[Reviews · NeurIPS 2017]

Reviewer 1



This propose proposes an expert-defined factor graph for analyzing EEG signal and inferring properties of brain areas, such as whether they belong to abnormal areas. The authors then develop a scalable inference algorithm for their model. They evaluate it empirically on real data and show it outperforms other methods. The paper is well written and clear. The role of G_n, the interaction between the channels, is unclear to me. It is mentioned throughout Section 3 yet I could not find it in the factor graph equations. The factor graph takes into account previous states and the authors claim to handle temporal dependencies. Yet, inference seems to be performed on a single epoch at a time and inferred values for previous states Y are never questioned again. Is there does a medical or biological motivation for this choice? From an inference point of view this looks problematic to me. As the states influence themselves over time, both past and future Fns would influence the states at any epoch n. Therefore it seems more logical to me to perform inference on all states together. Would that be possible with your method? Why not integrate the Maximum likelihood SOZ deduction within the model? The empirical results look good and show the qualities of the approach. I am however not very familiar with EEG analysis so I cannot comment on the interest of the proposed method for these applications. Suggestion: get rid of some lines in Table 2 so that you can have more space between the table and the following text. typos: l41: "Majority" --> A/The majority?

Reviewer 2



In the paper titled "EEG-GRAPH: A Factor Graph Based Model for Capturing Spatial, Temporal, and Observational Relationships in Electroencephalograms", the authors introduced a factor-graph-based model to capture the observational, temporal, and spatial dependencies observed in EEG-based brain activity data. From the methodology aspect, the techniques used in the paper is not new, but a combination of existing methods. The factor-graph model in the paper is quite similar to dynamic Bayesian networks. The MAP solution is achieved via min-cut algorithm. Note that, the graphical model is created manually, and therefore no structure learning is involved. From application aspect, this is a significant improvement in the fields of electrophysiology and brain computer interfaces because this is the first model which captures the three types of dependencies (claimed by the authors). The experiment design and evaluation is sound and solid. The proposed method significantly outperformed existing methods. The overall presentation is good and the paper is easy to follow. A few minor points: 1. It might be useful to mention whether the edge set E_n stays the same or not across different epochs. 2. What's the difference between the proposed method and dynamic Bayesian networks? It may be helpful to compare them. 3. The proposed method is also quite similar to a previous work (titled "High-Dimensional Structured Feature Screening Using Binary Markov Random Fields" by Liu et al, AISTATS-2012) which also used graphical model to capture dependence between variables, and employed min-cut algorithm to identify the binary hidden state (MAP inference). It may be helpful to compare them. 4. In Figure 1a, one arrow goes from "Current observation (events, rhythms)" to "Current state of a brain region". However, I feel the direction of the arrow should be reversed. What do you think?

Reviewer 3



SUMMARY: ======== The authors propose a probabilistic model and MAP inference for localizing seizure onset zones (SOZ) using intracranial EEG data. The proposed model captures spatial correlations across EEG channels as well as temporal correlations within a channel. The authors claim that modeling these correlations leads to improved predictions when compared to detection methods that ignore temporal and spatial dependency. PROS: ===== This is a fairly solid applications paper, well-written, well-motivated, and an interesting application. CONS: ===== The proof of Prop. 1 is not totally clear, for example the energy in Eq. (4) includes a penalty for label disagreement across channels, which is absent in the the graph cut energy provided by the proof. The relationship between min-cut/max-flow and submodular pairwise energies is well established, and the authors should cite this literature (e.g. Kolmogorov and Zabih, PAMI 2004). Note that the higher-order temporal consistency term can be decomposed into pairwise terms for every pair of temporal states. It is unclear why this approach is not a valid one for showing optimality of min-cut and the authors should include an explanation. What do the authors mean by "dynamic graph" (L:103)? Also, the edge set $E_n$ is indexed by epoch, suggesting graph structure is adjusted over time. It is not discussed anywhere how these edges are determined and whether they in fact change across epochs. This estimate of SOZ probability is motivated in Eq. (5) as an MLE. It isn't clear that (5) is a likelihood as it is not a function of the data, only the latent states. The estimated probability of a location being an SOZ is given as an average over states across epochs, which is a valid estimator, and connections to MLE are unclear beyond that. Generalizability claims of the approach (L:92-93) are weak. The extent to which this is a general model is simply that the model incorporates spatio-temporal dependencies. Specifying the factors encoding such dependencies in any factor graph will always require domain knowledge. Some detailed comments: * Are there really no free parameters in the model to scale energy weights? This seems surprising since the distance parameter in the spatial energy would need to be rescaled depending on the units used. * Authors claim no other work incorporates spatio-temporal EEG constraints but a recent paper suggests otherwise: Martinez-Vargas et al., "Improved localization of seizure onset zones using spatiotemporal constraints and time-varying source connectivity.", Frontiers in Neuroscience (2017). Please cite relevant material. * Lines in plots of Fig. 3 are too thin and do not appear on a printed version